# Eco-Sim: A Parametric Tool to Evaluate the Environmental and Economic Feasibility of Decentralized Energy Systems

**Karni Siraganyan, Amarasinghage Tharindu Dasun Perera** [ID]**, Jean-Louis Scartezzini** [ID] **and Dasaraden Mauree \*** [ID]

Solar Energy and Building Physics Laboratory, Ecole Polytechnique Fédérale de Lausanne, 1015 Lausanne, Switzerland; karnisira@gmail.com (K.S.); dasun.perera@epfl.ch (A.T.D.P.); jean-louis.scartezzini@epfl.ch (J.-L.S.)
**\*** Correspondence: dasaraden.mauree@gmail.com

**Abstract:** Due to climate change and the need to decrease the carbon footprint of urban areas, there is an increasing pressure to integrate renewable energy and other components in urban energy systems. Most of the models or available tools do not provide both an economic and environmental assessment of the energy systems and thus lead to the design of systems that are sub-optimal. A flexible and modular simulation tool, Eco-Sim, is thus developed in the current study to conduct a comprehensive techno-economic and environmental assessment of a distributed energy system considering different configuration scenarios. Subsequently, an intermodel comparison is conducted with the Hybrid Optimization Model for Electric Renewable (HOMER) Pro as well as with a state-of-the-art industrial tool. Eco-Sim is then extended by including the heating demand, thermal conversion (by using heat pumps and solar thermal) methods and thermal storage. A parametric analysis is conducted by considering different capacities of solar photovoltaics (PV), solar thermal panels and energy storage technologies. The levelized cost of electricity, the autonomy level and the $CO_2$ emissions are used as the key performance indicators. Based on the analysis of a study case conducted in a neighbourhood in Geneva, Switzerland, the study reveals that, with the present market prices for batteries and seasonal changes in solar energy potential, the combination of solar PV with battery storage doesn't bring a significant autonomy to the system and increases the $CO_2$ emissions of the system. However, the integration of thermal storage and solar thermal generation is shown to considerably increase the autonomy of the neighbourhood. Finally, multiple scenarios are also run in order to evaluate the sensitivity of economic parameters on the performance indicators of the system. Under the assumptions of the model, to foster investments in solar PV and battery installations, falling installation costs or stronger policies in favor of renewable energy seem necessary for the future.

**Keywords:** battery storage; electricity price; solar photovoltaic; solar energy; techno-economic assessment; thermal energy storage

---

## 1. Introduction

The extensive use of fossil fuels as the primary source of energy is the major source of greenhouse gas emissions and is hence responsible for the current climate change [1]. At the same time, the world population living in urban areas has since 2010 reached 50%, and this figure is expected to rise to 75% in 2050 [2]. It is thus urgent to address both these critical issues by decreasing our carbon based energy consumption and providing secure and sustainable sources of energy for the ever growing urban areas.

Renewable energy technologies are expected to play a major role in Switzerland to face such societal challenges such as climate change, resource depletion and the abandonment of nuclear energy by 2050 as mentioned by the law [3]. One of the possibilities of addressing these is by adopting the vision of a "2000-Watt Society" as developed by the Swiss Federal Institute of Technology (ETH) in Zürich. It is a model for energy policy which demonstrates how it is possible to consume only as much energy as worldwide energy reserves permit. This is possible when every person in every society limits their energy consumption to a maximum of 2000 watts [4] in 2050. The 2000-watt society's idea is to divide electrical and heating needs by four. Consequently, a fraction of the current energy consumption will still need to be provided by clean and affordable energy systems.

Decentralized energy systems combining different renewable energy technologies are becoming more popular to establish a more sustainable and autonomous power supply in neighbourhoods [5–9]. With the rapid improvement in energy efficiency and price reduction, solar photovoltaics (SPV) technologies are becoming promising [10,11]. Moreover, photovoltaics (PV) systems, for example, are scalable for applications ranging from watts to megawatts [12]. By 2050, it would be possible to meet around 20% of the current level of electricity demand in Switzerland through the use of photovoltaic systems only [13].

However, the stochastic nature of the resources is the main limitation to the installations of renewable energy systems [14]. Consequently, there are often gaps between consumption and the supply of the plants and there is a need to match the resource and demands that would encourage the use of renewable energy [12]. Energy storage is one solution to balance the fluctuations in demand and generation [15–17]. Furthermore, solar and battery technologies are often installed at individual building scale, and it could be interesting to look at such installation at neighbourhood scale [18]. This will clearly improve the potential of solar panels to be part of an integrated energy system [19].

A complete review of computational tools to analyze the integration of renewable energy systems was performed by Connolly et al. [20]. Previous studies highlighted the role of energy storage with renewable electricity generation [21] and the operation conditions of batteries in PV applications [22]. Other studies examined the economic viability of storage [23] and the use cases for stationary battery technologies [24]. Nonetheless, integrating storage technologies into solar PV systems increased the overall investment cost [25]. Some previously developed tools, such as SolarTherm [26] or the System Advisor Model (SAM) [26], have thoroughly addressed the economic assessment of renewable systems. It currently remains unclear when PV and storage investments will become economically and ecologically interesting in a large-scale application [27] in countries such as Switzerland with a low carbon mix [28]. The detailed assessment at neighbourhood scale have hardly been addressed to this extent [29,30] and often require extensive sources of data which are not easy to use. Multiple studies have recently analyzed the impact of using different temporal resolution for the load profile and for the generation of PV [31–34]. However, the simulation of whole districts to obtain load profiles for all the buildings in a district would require significant computational resources. Some studies have recently tried to addressed this issue with new simulation methods [35]. Additionally, methods have been developed in the literature addressing the dynamic behaviour of solar energy into the generation of electricity [36,37] and fuels [38], including the dynamic performance of power cycles [39] and the market of PV farms [40]. However, most of these methods rely on a coarse temporal resolution and thus an hourly temporal resolution is considered as a good compromise to capture both the dynamics of generation and consumption [15,41,42], although this does not guarantee an optimised outcome in the assessment and analysis of decentralized energy systems [32,43]. It is hence necessary to analyze the economic and ecological cost of a combination of different energy storage strategies [44,45] by combining multiple tools together as proposed by [46]. This requires a detailed simulation followed by a techno-economic and environmental assessment considering the lifetime operation of the energy system. Energy interactions within the system, cash flow and environmental impact at both design and operation stages should be considered at the neighbourhood scale.

A simulation tool, Eco-Sim, that aims to combine multiple energy conversion units technologies and to perform both an environmental and economical assessment has hence been developed. The objective of this work is to setup and validate a robust but practical simulation tool, and to evaluate the integration of solar energy and storage on a specific cluster of buildings in the Junction district of Geneva. The major aim of this study is to find the best configuration in terms of system size and to evaluate the life cycle emissions and cost of the system. Therefore, we will first estimate the economical and environmental cost of the solar PV panel, batteries, solar thermal and heat pumps, and we will then combine them to define an environmental and techno-economical cost model for an energy system in a neighbourhood in Geneva. Multiple scenarios will be studied with a combination of different energy storage size by changing the input parameters and initial assumptions to perform a sensitivity analysis and further analyze the future scenarios.

## 2. Methodology

A new model, Eco-Sim, was developed to include the technical, economical and environmental aspects in the assessment of the energy systems (see Figure 1). The approach chosen for the current study aims at considering the energy and cash flows for 8760 time steps in one year and over 20 years. As compared to previous studies, both the economical cost and environmental indicator ($CO_2$ emissions) will also be assesed in the current study. The inputs required for the model (energy demand, $CO_2$ emissions, cost for each technology) will be explained in the next subsections. Figure 2 shows the different components (the grid, solar photovoltaic panels, batteries, solar thermal, boilers and heat pumps) that are included in this study. Each of them will also be detailed in Section 3.5.

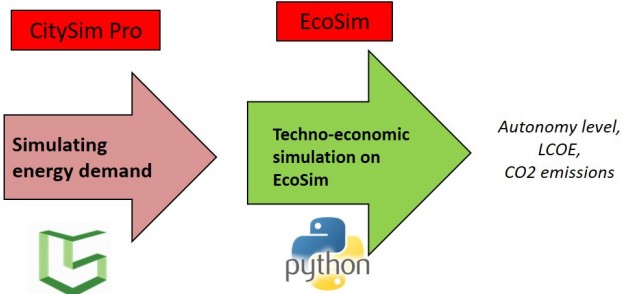

**Figure 1.** Flowchart describing Eco-Sim inputs and outputs.

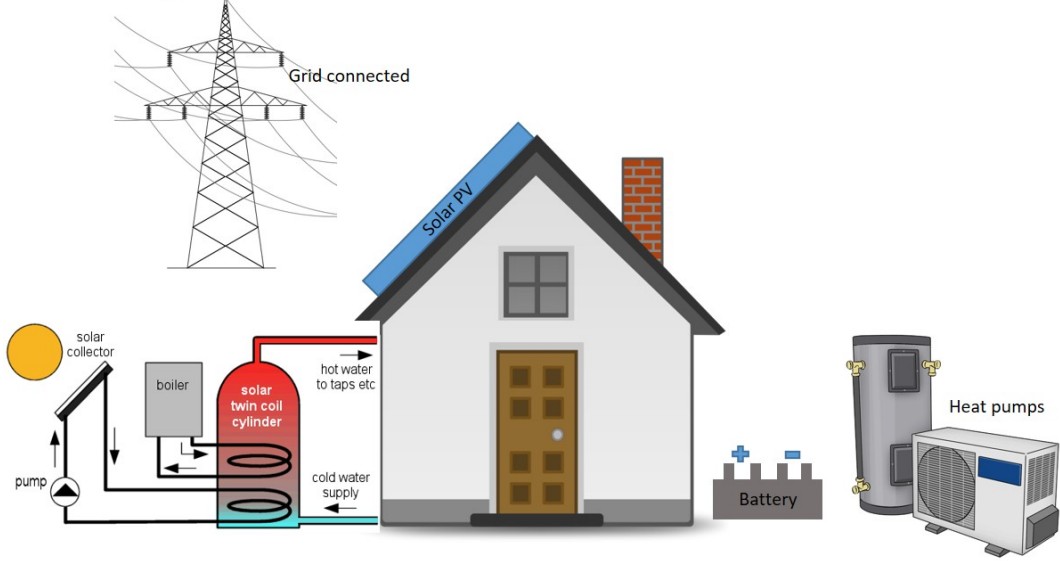

**Figure 2.** General scheme of various components of the proposed energy system.

*2.1. Techno-Economical and Environmental Assessment*

This section gives a brief overview of equations typically used in the techno-economic as well as environmental assessment of energy systems. Readers are encouraged to refer to [30,47,48] for a more detailed description. Three key performance indicators are calculated by Eco-Sim:

1.  The autonomy level: The autonomy level is defined as the share of generation by the energy system that is directly used by the consumers and is calculated by simulating the energy flows of the system over the year at an hourly resolution. It is thus assumed that whenever demand during the day meets the generation or available energy from the storage, the energy is consumed directly [23]. The ratio between electricity that is directly self-consumed and the total electricity demand defines the autonomy level [9,49] and thus gives an indicator of the dependence on the grid. It is calculated with the Equation (1):

$$Autonomy\ Level = \frac{\sum\limits_{t=0}^{8760} \left( D_t - E_{b_t} \right)}{\sum\limits_{t=0}^{8760} D_t} \tag{1}$$

2.  The Levelized Cost of Energy (LCOE) calculation: The net present value of a unit cost of electricity over the lifetime of the asset is the Levelized Cost of Energy (LCOE). All the costs relating to an electricity generating system are included in the LCOE and it is considered as a first order economic assessment criteria [50]. The total of the costs that are incurred during the lifetime of a technology is divided by the total energy demand and thus takes into account the fact that there will be differences in the lifetime of various technologies in the energy system [51]. The LCOE calculation is based on the following formula:

$$LCOE = -\frac{\sum\limits_{y=1}^{y=20} NPV}{20 \times \sum\limits_{t=0}^{8760} D_t} \tag{2}$$

The profitability of an system is computed by deducting the present values of cash outflows (including initial cost) from the present values of cash inflows over the $y$ years of lifetime of the system [52] and based on the investment over $y$ years. This leads to the net present value (NPV) as calculated with Equation (3). The investment costs, the operations, the maintenance expenses and the price of electricity bought from grid are all taken into account in the cash outflows while the price of electricity that is not self-consumed or stored, and is sold to grid represent the cash inflows. The NPV also include the replacement cost and the salvage cost of technologies that would become obsolete within the duration of $y$ years. This study aims at analyzing the energy system from the users's point of view.

$$NPV(y_1) = E_s - E_b - IC$$
$$NPV(y_{2-20}) = E_s - E_b - OM \tag{3}$$

The LCOE may vary from one technology to another depending on the number of factors. It can be highlighted from Equation (2) that a situation with a very high energy demand will lead to a smaller LCOE as compared to one with a lower energy demand. The LCOE gives with a useful metric to compare different costs of various technologies over the years and is hence

always computed as connected to the grid/network and is not calculated for a stand alone system (e.g., demand completely satisfied with only PV).

3.  $CO_2$ emissions: To calculate the $CO_2$ emissions of the whole energy system, each component has to be accounted for individually. To ensure that the $CO_2$ emissions of the complete life cycle of the technologies are taken into account, the specific $CO_2$ emissions based on the lifetime of the technology used and is computed as:

$$eCO_2 = \frac{\sum mCO_2 \times Q_c}{\sum\limits_{t=0}^{8760} D_t} \tag{4}$$

For example, if the demand is satisfied by energy from both the solar PV panels and from the grid, the $CO_2$ mix that will be calculated will include $CO_2$ emissions issued from the PV panels and also from the grid.

The equations given here are generalized ones which can be applied to any case study and in multiple other countries. In Section 3.5, more details will be given on the calculation of the LCOE and on the $CO_2$ emissions from each of the devices considered in the current study.

## 2.2. Dispatch Algorithm

The operating strategy for storage in the energy system is shown in Figure 3 and corresponds to previous strategies designed, for example in SolarTherm [53]. If the renewable generation is higher than the demand, the excess electricity can be stored in the battery. If the storage level has reached the maximum limit, the excess electricity can be sold to the grid. Similarly, if the demand is higher than the generation, two options are available: Buy from the grid or use the available stored electricity in the battery. The minimum state of charge is defined as a threshold such that the energy stored in the battery cannot be used [54].

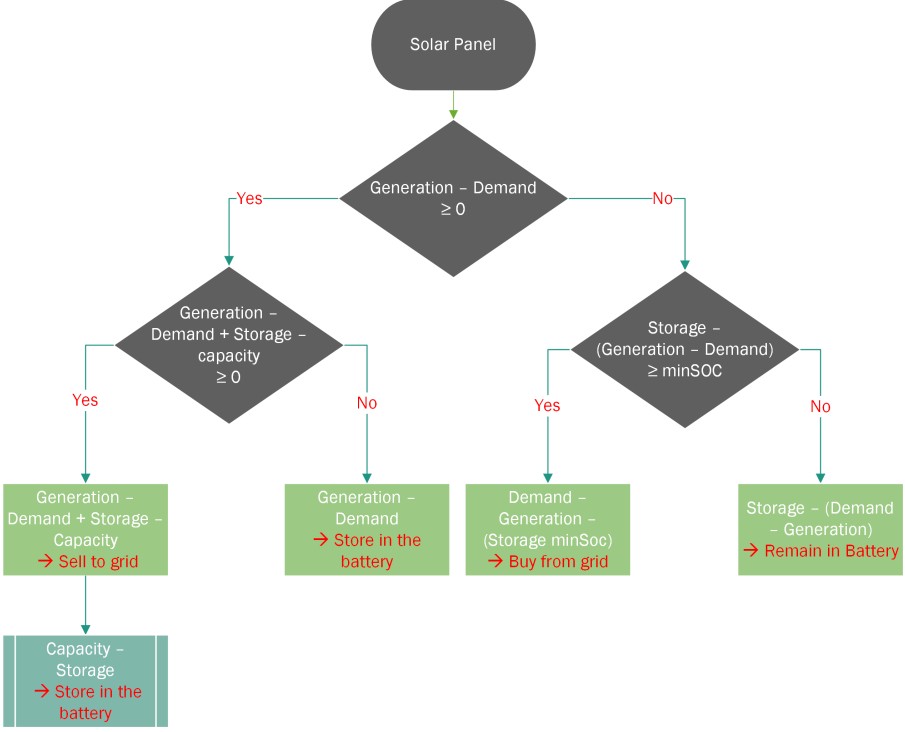

**Figure 3.** Strategy for the battery operations.

### 2.3. Sizing of the Storage

Currently, the sizing of the storage is usually done with respect to the demand. The advantage of using such a methodology is the optimization of the size of the batteries based on the demand. However, the drawback to this is that the energy produced is not often consumed locally and hence the full potential of the generation of the system is not used [55,56]. In order to address this, we propose here to determine the size of the storage by using an average day, representative of a typical day over the whole year and by integrating over time. This methodology will be further used to calculate the storage capacity of the battery and also the thermal storage capacity in Section 3.4.

## 3. Description of the Case Study and of the System Design

### 3.1. Junction District in Geneva

A neighbourhood containing approximately 800 buildings in the Junction district in Geneva, Switzerland is considered for this study. In a previous study, the electricity demand, the heating demand and the solar generation of each building were simulated using CitySim [57] for every hour (8760 time steps) over one typical meteorological year [58] using the Meteonorm dataset [59]. Depending on the location and occupations, some buildings had a higher electricity demand than other buildings. Regrouping buildings at such a large scale could thus be advantageous to smoothen the load profiles if there is a variety of occupancy profiles.

### 3.2. General Assumptions

Since we are modelling the Junction district of Geneva, we choose the Swiss franc (CHF) as the currency (note that 1CHF is currently $1). Naturally, all the houses are connected to the grid to cover the demand not supplied by the local resources. Based on a data analysis of consumer prices, the average price of electricity bought over the network in Geneva in 2016 is 20.6 cts/kWh [60] and the average selling price is 10.9 cts/kWh [60]. The price of electricity in Geneva has decreased by 7% compared to 2000 prices but has increased by 2% per year over the past two years [60]. In our model, we will however assume the general consensus that the electricity and the selling price increases every year by 2%. Additionally, a discount rate of 2% is also used over the period of 20 years. We will also assume that the natural gas buying and selling price increases by 2% each year. The $CO_2$ emissions of the electrical grid in Switzerland is 0.155 kg/kWh. The $CO_2$ emissions of the electrical grid in Switzerland is determined based on hourly values of the electricity taken on the network during a specific day [61]. The $CO_2$ emissions of the electrical and natural gas grid in Switzerland are 0.160 kg/kWh by considering that in 2015 the energetic mix in Geneva was composed by 5% of natural gas [62].

### 3.3. Energy Generation and Demand

The PV electricity generation in kWh is obtained by using the available irradiation (obtained from the CitySim software) in hourly resolution for a specific cluster in the Junction District of Geneva. In a different study, we used the CitySim software to evaluate the generation from installed PV on top of individual buildings [58]. Since this is outside the scope of this paper, as we are working at the neighbourhood level, we only consider inefficiencies in the PV system, such as inversion losses, to calculate the solar PV generation and hence the irradiation is multiplied with a solar cell efficiency of 15% [63]. The maximum capacity of solar PV installed is finally determined based on the peak generation of the cluster. (Note that the maximum hourly irradiation over the year in Junction District is 1094 kWh/m$^2$ [64]).

Figure 4a illustrates the electricity generation and demand over a year. As can be expected the demand is higher compared to the electricity generated. Figure 5 illustrates the generation divided by the demand during different seasons. Clearly during the summer time, when the generation is higher than the demand, the excess energy can be stored. However, it can be seen that when working with a

time resolution of an hour, there are significant other periods during the year where the generation is higher than the demand.

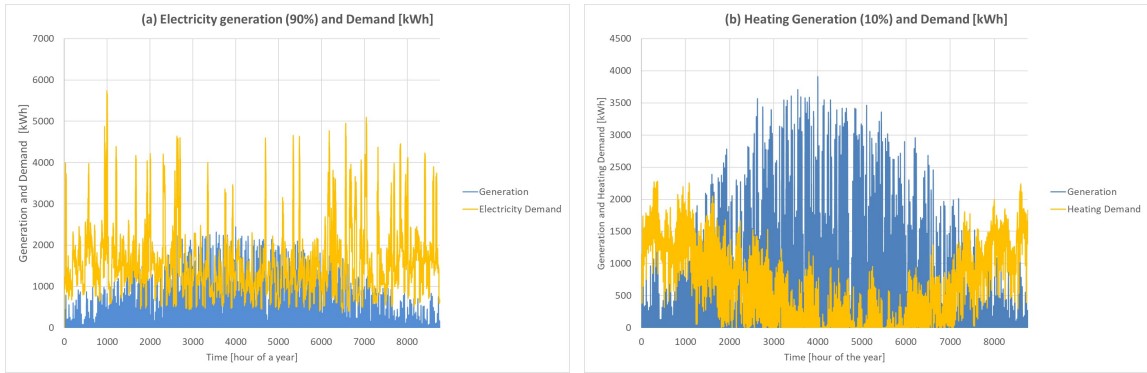

**Figure 4.** (**a**) Photovoltaics (PV) electricity generation (example with a solar PV capacity of 2197.7 kWp) and electricity demand over a year; (**b**) Solar thermal generation and heating demand over a year (example with a solar thermal capacity of 1302.34 kWp).

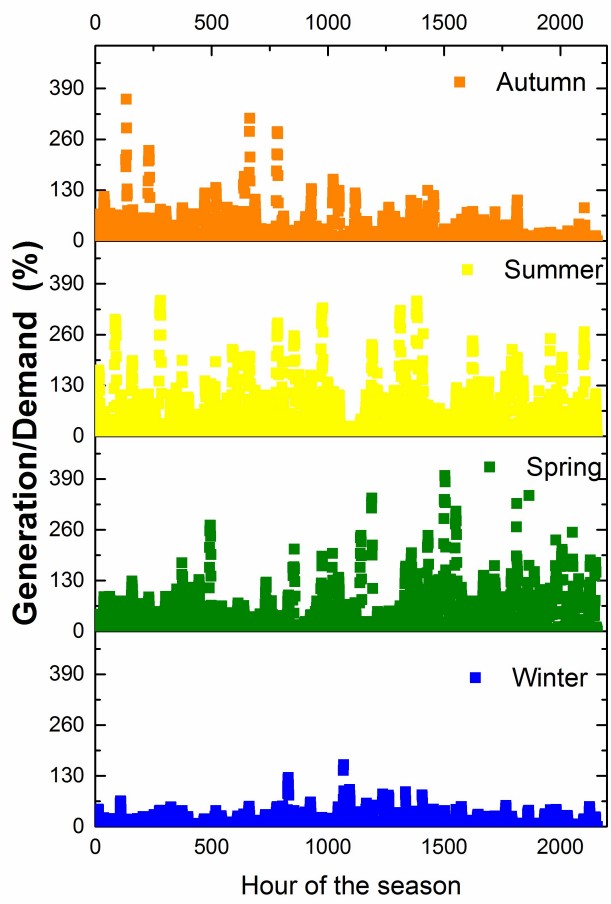

**Figure 5.** Solar PV generation (90% scenario with a capacity of solar PV of 2197.7 kWp) divided by the electric demand for different seasons.

The solar thermal generation in kWh is obtained by using the available irradiation in hourly resolution for a specific cluster in the Junction District of Geneva. To reflect inefficiencies in the thermal grid, the solar thermal generation is multiplied with an efficiency of 80% [65]. Figure 4b illustrates the generation and heating demand over a year. The mismatch between generation and heating demand

can be stored in a boiler to be used when it is required. If the boiler capacity is reached, the excess energy produced during summer can be sold to the grid. The efficiency of the boiler is set to 93% [63].

### 3.4. Characterization of the System

The capacity of storage is determined based on the Figure 6 as the integral below the bold blue line (which is the average generation of the system). This calculation leads to a battery capacity of 7582.9 kWh. As mentioned previously, the capacity of storage is only based on the average generation in order to maximize the use of the solar potential.

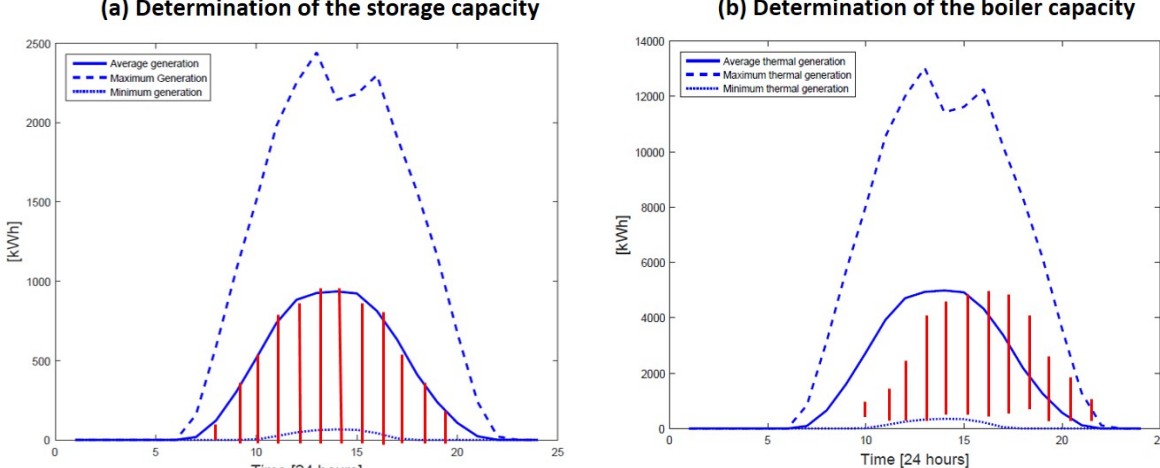

**Figure 6.** (**a**) Determination of the electrical storage capacity from the energy generation of the system (maximum, minimum and average); (**b**) determination of the boiler capacity from the energy generation of the system (maximum, minimum and average).

The capacity of boiler is determined based on the Figure 6. The calculation leads to the boiler capacity of 40,442 kWh, using

$$Q = mc_p\Delta\theta \tag{5}$$

The mass of water can be found with using $c_p$ = 4.2 kJ/kg/K, $\Delta\theta$ = 60 K (considering an application with water at 70 K) and $Q$ = 145,591,200 kJ, which leads to a boiler of capacity of 578,000 kg or 578 m$^3$. The maximum capacity for each of the considered technologies are presented in the Table 1.

**Table 1.** Energy technologies and the maximum capacity for each technology.

| Technology | Capacity |
|---|---|
| Solar PV panel | 2441 kW |
| Battery | 7582.9 kWh |
| Solar thermal panel | 13,023.41 kW |
| Boilers | 578 m$^3$ |
| Heat Pump | 4070 kW |

### 3.5. Energy System Configuration Scenarios

Different technologies are considered for the energy system: Solar PV, solar thermal, storage technologies (lead–acid or lithium-ion batteries) and also assisting energy technologies such as boilers and heat pumps. A combination of different components have been tested with multiple scenarios by changing their maximum installed capacity. In the next section, a detailed description of the four different scenarios are presented based on the input parameters that have been used in the model. An example scenario tree including all the combinations that we considered for solar PV and battery is given in Figure 7 (Scenarios 3 and 4 are not presented in the scenario tree).

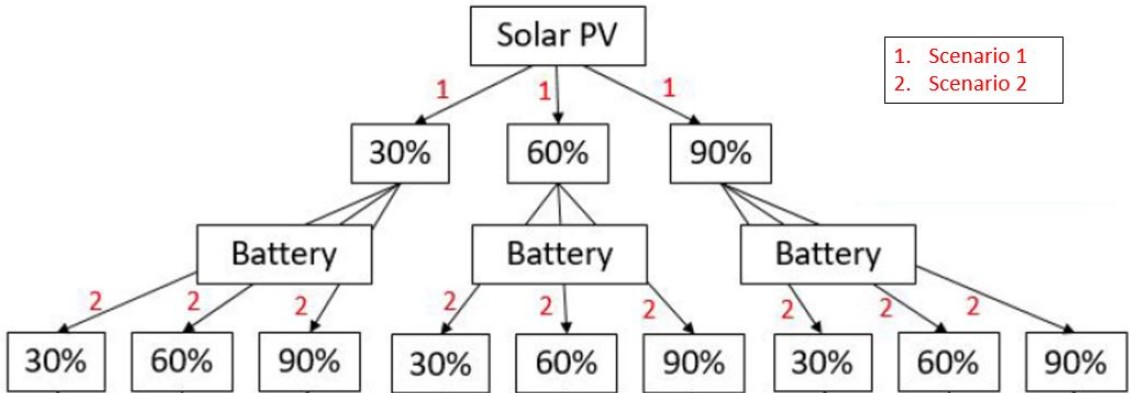

**Figure 7.** An example scenario tree including all the combinations that we considered for solar PV and battery (values at the top is for the first scenario and values below is for the second scenario including solar PV and battery).

- **Scenario 1: Solar Panels.** In the first scenario, 30%, 60% and 90% of the maximum capacity of solar panel are considered. The input parameters for large scale PV system are summarized in Table 2. The LCOE of PV by itself is calculated by taking into account all the cost during the lifetime of the technology such as the given investment, the O&M cost and the discount rates. The cost per kW of solar PV is given in Table 2. The entire PV system life cycle (transport, operation, electric installation, construction and production phase) is considered to calculate the $CO_2$ emissions for solar panels. Previous studies have also obtained similar values when computing the entire $CO_2$ emissions for a PV system [66,67].

- **Scenario 2: Solar Panels and batteries.** The second scenario combines different percentages of the maximum possible installation capacity of solar panels (30%, 60%, 90%) and batteries (30%, 60%, 90%). Two types of storage technology are compared in this study: Lead–acid battery and lithium-ion battery. The input parameters for large scale batteries are described in Table 3. The LCOE of PV and battery system by itself is calculated by taking into account all the costs during the lifetime of those technologies such as the given investment, the O&M cost for solar PV and batteries and their discount rates. For the year 20, we added the salvage value of a used battery (1/3 of the battery price after 20 years). We assumed that the battery price decreases each year by 7.6%. The energy efficiency reflects the losses during charging and discharging periods. The characteristics of these two types of battery are summarized in Table 3 [68]. Note that we have not accounted for battery inverters in these scenarios.

**Table 2.** Economic input parameters for solar PV.

| Solar Panel | Unit | Value | Sources |
|---|---|---|---|
| Cost | [CHF/kW] | 1350 | [69] |
| Lifetime | [years] | 20 | [51] |
| O&M cost | [CHF] | 1.5% of SP cost per year | [70] |
| $CO_2$ emissions | [kg/kWh] | 0.044 | [71,72] |
| Electrical grid $CO_2$ emissions over a year without PV | [kg/kWh] | 0.155 | [61] |
| Electrical and natural gas grid $CO_2$ emissions over a year without PV | [kg/kWh] | 0.160 | [61] |

- **Scenario 3: Solar Panels and heat pumps.** The third scenario combines different percentages of solar panels (30%, 60%, and 90%) as in the second scenario but in this case the electricity demand of the heat pump is added to satisfy the heating demand. We assume that the heat pump has a constant COP of 3.2, although the efficiency of heat pumps are subject to climatic conditions. The hourly heating demand is divided by the COP and is added to the current electricity demand.

The LCOE of the solar PV and heat pump by itself is calculated by taking into account all the cost during the lifetime of those technologies (investment, the O&M cost, the discount rates).

- **Scenario 4: Solar Panels, heat pump and solar thermal.** The fourth scenario combines different percentages of solar panels (30%, 50%, 70%) and solar thermal panels (70%, 50%, 30%) that corresponds to the remaining available roof area and also heat pumps. In this scenario, the aim is to satisfy both the heating demand and the electricity demand. The input parameters for solar thermal panels and boilers are described in Table 4. The heating demand that is not totally satisfied by using solar thermal panel is satisfied by using heat pumps. The whole demand is then satisfied by using solar thermal panels, heat pump and solar PV. We assume here that a district heating network is installed (as proposed in the current construction plan of the district) and that excess heat produced in the district can be sold to the grid (at 6.3 cts/kWh).

**Table 3.** Economic input parameters for batteries.

| LEA Battery | Unit | Value | Sources |
|---|---|---|---|
| Cost | [CHF/kWh] | 163 | [73] |
| O&M cost | [CHF] | 22 | [70] |
| $CO_2$ emissions | [kg/kWh] | 15 | [74] |
| Lifetime | [years] | 7 | [74] |
| Efficiency | [%] | 81 | [74] |
| MSC | [%] | 30 | [74] |
| Energy density | [Wh/L] | 100 | [74] |
| **LI-Ion Battery** | **Unit** | **Value** | **Sources** |
| Cost | [CHF/kWh] | 440 | [73] |
| O&M cost | [CHF] | 19 | [70] |
| $CO_2$ emissions | [kg/kWh] | 70 | [74] |
| Lifetime | [years] | 15 | [74] |
| Efficiency | [%] | 92 | [74] |
| MSC | [%] | 20 | [74] |
| Energy density | [Wh/L] | 250 | [74] |

**Table 4.** Economic input parameters for thermal demand.

| Heat Pump | Unit | Value | Sources |
|---|---|---|---|
| Cost [VITOCAL 300-G] | [CHF/kW] | 1326.54 | [75] |
| Lifetime | [years] | 15 | [76] |
| $CO_2$ emissions | [kg/kWh] | 0.04 | [77] |
| **Solar Thermal Grid** | **Unit** | **Value** | **Sources** |
| Cost [Vitosol 200-F] | [CHF/m2] | 432.6 | [75] |
| Lifetime | [years] | 20 | [78] |
| OM | [CHF] | 0.58 % of IC | [79] |
| $CO_2$ emissions | [kg/m$^2$] | 0.040 | [80] |
| **Natural Gas Boiler** | **Unit** | **Value** | **Sources** |
| Cost [Vitocell 100-L] | [CHF/m$^3$] | 4564.5 | [75] |
| Lifetime | [years] | 15 | [81] |
| $CO_2$ emissions | [kg/m$^3$] | 0.225 | [82] |

## 4. Results and Discussions

### 4.1. Model Validation

In this section, an intermodel comparison is performed for the PV system and batteries. A comparison is made between the two first scenarios by only considering the electricity demand. The best scenarios were those that have the lowest LCOE value to be competitive with the grid price.

In addition, we also try to choose the scenario that maximized the autonomy level and minimized the $CO_2$ emissions to have an energy system that is more autonomous and sustainable.

### 4.1.1. Solar PV Panels

As expected, the autonomy level increased with the percentage of solar panel that we installed. An autonomy level up to 16% could be reached by using 90% solar panels. It can be noted that the $CO_2$ emissions decreased considerably (0.138 kg/kWh) with 90% rooftop solar PV as compared to the $CO_2$ emissions of the electrical grid in Switzerland (0.155 kg/kWh) which corresponded with previous studies conducted in Switzerland [83]. If we focus only on the economic competitiveness of the system, the best scenarios were those that minimized the LCOE value to be competitive with the grid price of 0.206 CHF/kWh. Nevertheless, it could be seen that the 90% scenario with a capacity of 2.2 MW still had a higher LCOE value (see Table 5). In addition, by considering the investment cost as well as the O&M cost of solar PV, we obtained the LCOE of solar PV as a stand alone system. Figure 8 gives an example of the net present value calculation for solar PV. As mentioned in Section 2, the NPV was calculated from the perspective of the consumer (meaning positive for selling electricity and negative for buying electricity), over the lifetime of the proposed system of over 20 years considering that the investment was made in year one and with a discount rate of 2%.

**Table 5.** Results for solar PV (Scenario 1).

| SP | LCOE [CHF/kWh] | Autonomy Level | $CO_2$ Emissions [kg/kWh] |
|---|---|---|---|
| 30% | 0.238 | 6% | 0.148 |
| 60% | 0.227 | 11% | 0.143 |
| 90% | 0.217 | 16% | 0.138 |

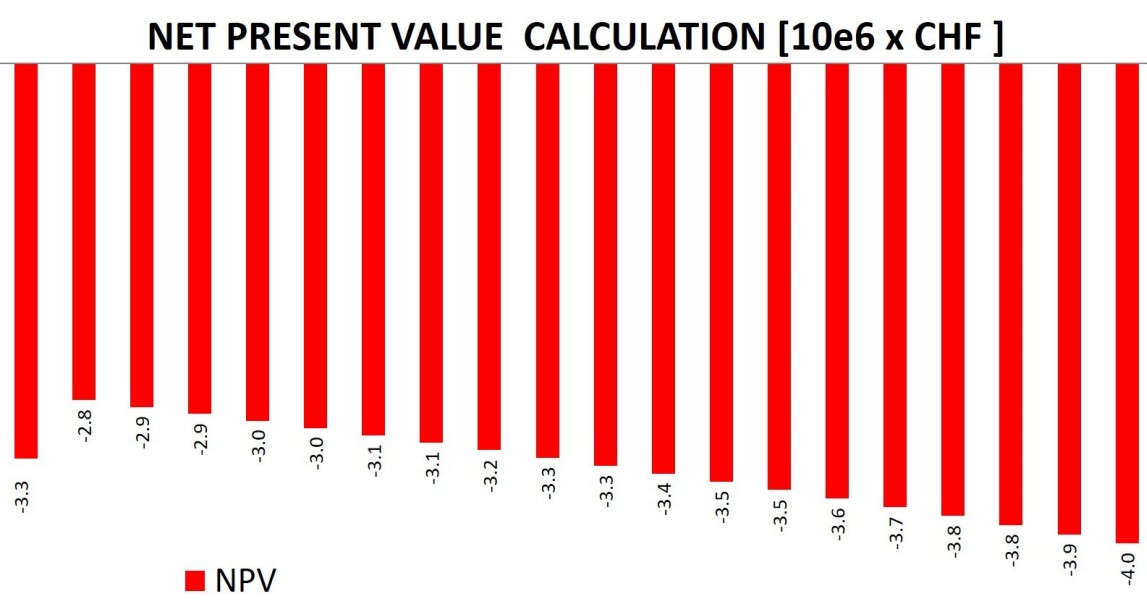

**Figure 8.** Net present value calculation for solar PV over 20 years.

We compared our model with the Hybrid Optimization Model for Electric Renewable (HOMER) Pro software [84] for the LCOE and with an industrial tool. HOMER Pro is a micropower optimization model that mostly evaluates the available energy technologies "to identify the least-cost solutions to energy requirements" [84]. This was done to test the robustness of the economic solutions obtained for the solar PV panel.

Multiple configurations of Eco-Sim were compared with outputs from the HOMER Pro software. It should be noted for example that there are some differences in the assumptions made in the two software. For example, HOMER Pro software uses a normalized value for "generic flat plate PV" for Geneva and has a total generation of 1,873,735 kWh/yr which leads to a production 1.13 times higher than our total generation. Additionally, the inflation rate is considered as 1% in HOMER Pro while it is of 2% in our model. To test the performance of Eco-Sim, the inflation rate was brought to 1% and the production was also modified to correspond to the HOMER Pro's. When the same assumptions were used in both models, Eco-Sim gave an estimation of the LCOE that is equal to the one given in HOMER Pro software (e.g., for the 60% case an LCOE of 0.204 CHF/kWh is obtained).

We then also compared the results for the autonomy level with a state-of-the-art industrial tool "BARTPower" from the [85]. This comparison was done under a specific license and confidential agreement with the industrial partner. "BARTPower" performs similar tasks as HOMER Pro although it has a limited number of possible energy systems and configurations. A comparison was, for example, conducted for the 60% rooftop solar panel. There was no difference in the autonomy level (13%) that was obtained for both models. Figure 9 shows an example of an hourly profile of the self-consumption obtained from Eco-Sim.

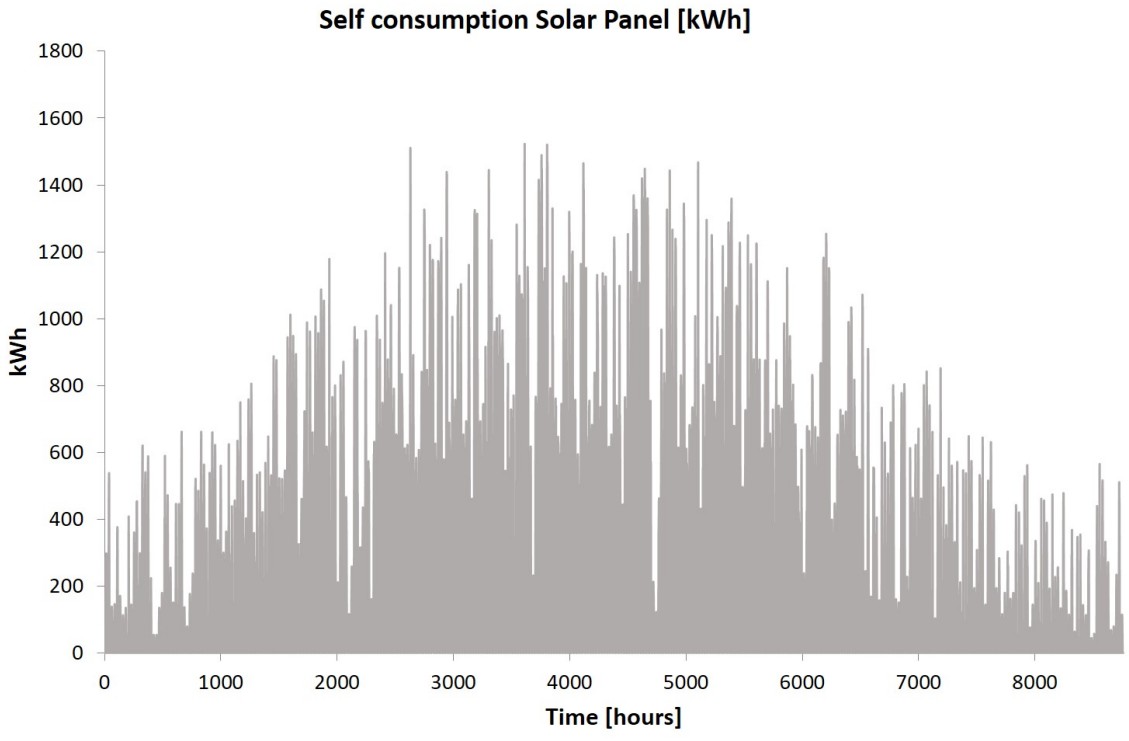

**Figure 9.** Autonomy level (self-consumption) over a year for 30% scenario.

4.1.2. Solar Panels and Batteries

For these scenarios, our findings show that battery technology doesn't bring any additional autonomy level for the considered electricity generation of our study case. Battery technologies were still expensive and added considerable additional cost to the PV system. For instance, the lowest LCOE that we obtained with a battery was 0.222 and with a lithium-ion battery was 0.225 with 90% of solar PV and 30% of battery. The main drivers behind this variation of LCOE were the difference in investment cost between those two technologies. It seems that the integration of battery technology was still not profitable economically for large scale installation. A lower battery size of 30% was more adapted to reduce the LCOE. Moreover, a decrease in battery cost could allow a lower LCOE and

could lead to a high economic viability of storage. In addition, by considering the investment cost as well as the O&M cost of battery, we obtained the LCOE of battery as a stand alone system.

In a second step, we compared the results for solar PV and batteries with the simulation tool called "BARTHome" from SI-REN. "BARTHome" is an extended version of "BARTPower" but also evaluates the integration of batteries. It should however be noted that in "BARTHome", it is also possible to define the orientation of the solar PV panels. A very good agreement between Eco-Sim and "BARTHome" was obtained in the case of the autonomy level (for example 17% for the 90% rooftop PV and with 30% lead-acid batteries). The readers can refer to Siraganyan et al. [30] for further details.

### 4.2. Extension of the Model with Thermal Demand

In the following section, the validated simulation tool is extended to include the thermal demand as well as the electricity demand. A comparison is made between the third and fourth scenario by considering the electricity demand as well as the thermal demand.

#### 4.2.1. Solar Panels and Heat Pump

Battery technologies are not used in this section because the previous findings demonstrate that batteries are not well adapted in a large scale application. Table 6 shows that the lowest LCOE that we obtained was 0.224 with 90% of solar PV. The heating demand as well as the electricity demand was satisfied in this scenario. The $CO_2$ emissions were again relatively lower than the grid since the system was now providing at least 14% of the demand as can be noted from the autonomy level figures. A higher autonomy level could have been reached if the hours of operation of the heat pump were in better correspondance with the production from the PV. This could be achieved for example by using real-time pricing to encourage use of locally produced energy during peak time hours.

**Table 6.** Results for scenario 3 with solar panels and heat pump.

| SP | LCOE [CHF/kWh] | Autonomy Level | $CO_2$ Emissions [kg/kWh] |
|---|---|---|---|
| 30% | 0.264 | 5% | 0.150 |
| 60% | 0.243 | 10% | 0.145 |
| 90% | 0.224 | 14% | 0.141 |

#### 4.2.2. Solar Panels, Heat Pump and Solar Thermal

The combination of 90% of solar panel and 10% of solar thermal with heat pumps gave an LCOE value of 0.222 with an autonomy level of up to 14%. The autonomy level wasn't maximized by adding solar thermal panel compared to the previous scenario with only solar PV and heat pump but the obtained LCOE values were much lower. Solar thermal allowed an economically more advantageous possibility compared to the case where we only had heat pumps and solar PV. This system is an attractive process to generate both heating and electrical energy simultaneously by using solar thermal, heat pumps and solar PV panels. It offsets 0.01 $CO_2$ emissions per kWh with 90% solar PV and 10% solar thermal that was slightly higher than the Scenario 3 (see Table 7).

**Table 7.** Results for solar panels, heat pump and solar thermal.

| SP | ST | LCOE [CHF/kWh] | Autonomy Level | $CO_2$ Emissions [kg/kWh] |
|---|---|---|---|---|
| 30% | 70% | 0.236 | 6% | 0.150 |
| 60% | 40% | 0.230 | 11% | 0.144 |
| 90% | 10% | 0.222 | 14% | 0.141 |

### 4.3. Cost Sensitivity Analysis and Future Scenarios

Finally, we present a sensitivity analysis by varying the input parameters. We extend our analysis to show the LCOE changes when varying the most critical input parameters that are covered by

Scenario 1 and Scenario 2 with lithium-ion battery. To analyze how the Scenarios 1 and 2 respond to a change of parameters first, we have varied the grid price and the selling price by −10%, −20%, +10% and +20%. As can be seen from Figure 10, the model was very sensitive to the variation of the grid price. A lower LCOE value was obtained by decreasing the grid price up to 20% and increasing the selling price up to 20%.

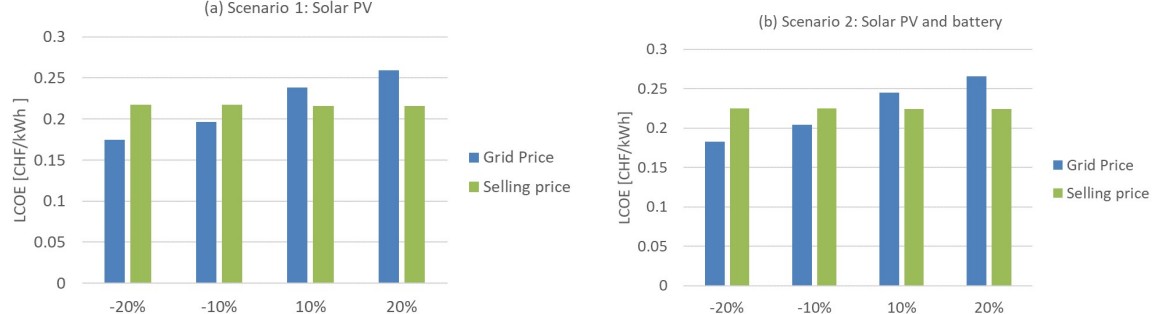

**Figure 10.** (**a**) Scenario 1 by changing the grid price and selling price; (**b**) Scenario 2 by changing the grid price and selling price.

Current trends in Switzerland assume a decrease in electricity prices for 2017 and a concurrent increase in the selling prices that can lead to a high economic profitability for solar PV and battery scenarios.

The model is sensitive to changes in the assumption of future battery and solar PV cost decrease. The future scenarios were conceived by reducing the solar PV and battery investment costs. The cost for solar PV and battery is expected to decrease by 10% in 2020 and by 20% in 2025 respectively, according to [86]. If this is applied, then the following can be noted:

1. A decrease of 1% is observed between 2016 and 2025 in the LCOE value of the first scenario by only changing the solar PV price.
2. A decrease of 1% is observed between 2016 and 2025 in the LCOE value of the second scenario by only changing the battery price.
3. By changing the solar PV price as well as the battery price, the LCOE value decreases by 2% between 2016 and 2025.

It becomes obvious that of all input parameters, the grid price, the battery and solar PV investment cost reduction have the greatest effect on the model.

The LCOE value fluctuates by the change of the solar PV and battery price (Figure 11). However, the obtained values for solar PV and battery are still not under the grid price that is 0.206. To be under the grid price, the LCOE value should decrease by 9% between 2016 and 2025.

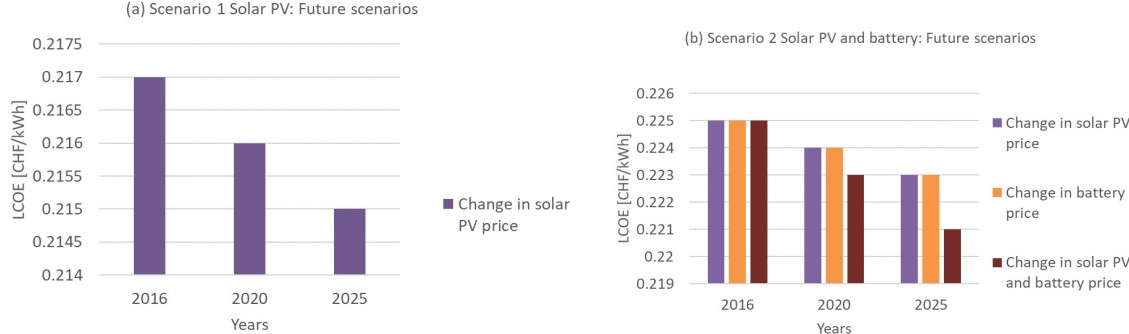

**Figure 11.** (**a**) Levelized Cost of Energy (LCOE) in the "breakthrough" scenario 1 in 2020 with a decrease in the investment cost of solar PV by 10% and with a decrease by 20% in 2025; (**b**) LCOE in the "breakthrough" scenario 2 in 2020 and 2025 with a decrease by 10% and 20% respectively in the investment cost of solar PV and battery price.

### 4.4. Implications on the Environment

Besides economic considerations, the adoption of PV, battery and solar thermal technologies depends on environmental factors. Solar PV system generates relatively low $CO_2$ emissions but the increase of battery diffusion in the markets raises environmental pollution significantly. Lead–acid batteries contain sulfuric acid and are toxic and generate carbon emissions. These environmental impacts can be reduced by recycling the lead–acid. The batteries can be charged many times, but after numerous uses, lead–acid plates deteriorate and the battery loses its efficiency. Lithium-ion batteries contain, among others, useful metals as cooper and aluminum as well as transition metals. Recycling processes for lithium batteries are also needed to enable a sustainable life cycle of these technologies. In Switzerland, nearly 70% of the commercial batteries were recycled in 2012 [87]. Combining 90% solar PV, 10% solar thermal with a heat pump gives a total $CO_2$ emissions of 0.141 kg/kWh that is relatively lower than the case without the use of those technologies, which is 0.155 kg/kWh. The results for the autonomy level and the penetration of renewable energy in the system is in agreement with previous studies that have been conducted in Switzerland [56,83].

The limitation of the thermal storage is the large water tanks because they could have corrosion and fouling. Several studies have looked at the performance of new possibilities for storages. For example, Liu et al. [88] looked at multiple technologies (phase-change materials, compressed air, etc.) that could be integrated in buildings, in particular in net-zero energy buildings. A review of thermal insulating materials that would increase the efficiency of themal storage was also recently proposed [89]. Other methods to increase the efficiency of the system were also analyzed by Nourozi et al. [90]. Storage, although still being a hurdle, is thus rapidly changing and future prospects such as storing heat in buildings, in the ground or subsurface aquifer could be another solution to store proper thermal energy [74,91]. Although heat pump systems generate low carbon emission, combining solar PV and heat pump is not critical regarding the environmental impacts according to our findings but can be used as a means to completely satisfy demand.

### 4.5. 2000-Watt Society

In a future scenario, with the 2000-watt society it would be possible to divide by four the electrical demand in 2050. This would thus lead to an autonomy level of 34% that could be reached with a rooftop fraction of 90% solar PV with a relatively low LCOE value and $CO_2$ emissions of 0.151 CHF/kWh and 0.129 kg/kWh respectively (Table 8). In addition, the findings show that battery technology (lead–acid battery or lithium-ion battery) can bring an additional autonomy level for the considered electricity generation of our study case if we considered the 2000-watt society's case as can be seen in Table 9. An autonomy level up to 52% could be reached by adding a 90% lead-acid battery with 90% solar

PV. The $CO_2$ emissions however increased as expected but the LCOE of 0.200 CHF/kWh remained competitive with the grid price (0.206 CHF/kWh).

**Table 8.** Results for solar PV (scenario 1) in a 2000-watt society.

| SP | LCOE [CHF/kWh] | Autonomy Level | $CO_2$ Emissions [kg/kWh] |
|----|----------------|----------------|----------------------------|
| 30% | 0.207 | 20% | 0.134 |
| 60% | 0.177 | 29% | 0.129 |
| 90% | 0.151 | 34% | 0.129 |

**Table 9.** Results for solar PV and lead-acid battery in a 2000-watt society.

| SP | LEA | LCOE [CHF/kWh] | Autonomy Level | $CO_2$ Emissions [kg/kW] |
|----|-----|----------------|----------------|--------------------------|
| 90% | 30% | 0.164 | 44% | 0.143 |
| 90% | 60% | 0.180 | 50% | 0.161 |
| 90% | 90% | 0.200 | 52% | 0.186 |

Finally, to satisfy both thermal and electrical energy, the heating demand was reduced by four in a future scenario. By adding solar panels (90%), heat pumps and solar thermal panels (10%), an autonomy level of 16% could be achieved with an LCOE value of 0.218 CHF/kWh and low $CO_2$ emissions (0.133 kg/kWh) (see Table 10). After extending the model with thermal and electrical energy for a future scenario, the findings show that adding solar PV (90%) with 10% solar thermal and heat pump would achieve significantly low $CO_2$ emissions and a competitive LCOE value.

**Table 10.** Results for solar panels, heat pump and solar thermal in a 2000-watt society.

| SP | ST | LCOE [CHF/kWh] | Autonomy Level | $CO_2$ Emissions [kg/kWh] |
|----|-----|----------------|----------------|----------------------------|
| 30% | 70% | 0.224 | 6% | 0.154 |
| 60% | 40% | 0.221 | 11% | 0.153 |
| 90% | 10% | 0.218 | 16% | 0.133 |

## 5. Conclusions and Perspectives

The current study aims to present a decision-making support tool for urban planners, energy system designer and researchers by reviewing costs and environmental impacts of solar PV, battery technologies, solar thermal panels and heat pumps. In this paper we devise a practical parametric simulation tool, Eco-Sim, that can be used at an early design stage, to investigate the techno-economic assessment at an hourly time step throughout a typical year.

Key performance indicators such as the LCOE, the autonomy level or the $CO_2$ emissions are given as an output of Eco-Sim. The tool is flexible and modular and thus can be easily extended or adapted to other uses as it has been demonstrated in the current study.

The study was conducted in a systematic way to build the tool progressively and to demonstrate the robustness of each module. An intermodel comparison was conducted using the HOMER Pro software as well as a state-of-the-art industrial tool from SI-REN.

The results obtained from Eco-Sim for a case study conducted over a neighbourhood in Geneva were in very good agreement with both models. According to the simulations for this neighbourhood in Geneva, our findings suggest that there were currently no significant advantage to combine solar PV with batteries. The Eco-Sim model was then extended to investigate the impact of including solar thermal panel and heat pumps in the energy mix of the neighbourhood in order to satisfy both the electrical and the heating demand. We found that combining solar PV, solar thermal and heat pumps is economically more advantageous than the combination of only solar PV and heat pumps. Combining 90% solar PV panel, 10% solar thermal with heat pumps gives the best and lowest LCOE value to

satisfy both the thermal and electrical demand. This combination allows a reduction of 10% of the $CO_2$ emissions as compared to the current emissions.

A few factors can help the integration of solar PV. A reduction in the price gap between buying and selling prices would increase the penetration of such systems. Moreover, a decrease in the solar PV price allows a decrease of the LCOE by 1%. Current trends in Switzerland assume a decrease by 4% in PV installation cost in 2017 that can lead to a lower LCOE of 0.216 instead of 0.217. Given that the market prices have fluctuated significantly over the years, future scenarios are difficult to predict.

We also found that a decrease in electricity price and a concurrent increase in the selling price can increase the demand of solar PV and battery installation. We concluded that, under the assumptions of our model, to foster investments in solar PV and battery installations, a decrease in the costs of investments seems necessary for the future. In addition, a final scenario demonstrated the impact of the 2000-watt society.

There were several limitations regarding the generation using PV systems in this study. The orientation and the technology of solar PV panels are not considered in the model. This work can be improved by adding the orientation of panels or by working with a specific efficiency for different photovoltaics technologies or taking production values directly from well known softwares such as CitySim [57] or PVSyst. Although we have only considered a constant COP for the heat pumps, the modeling framework can be slightly modified to account for changing efficiencies in the future. On another hand, the limitation of solar PV and storage system is that we restricted the choice of battery technologies to two different types of battery. In addition, we considered an average electricity price in Geneva but Eco-Sim can also be extended to use as input varying energy prices. This could help for example decision-makers on the need for varying feed-in tariff which can lead to load shifting and hence be more advantageous for integration of renewable energies. Additionally, storage can also have an implication on the economic value with varying prices. Our work has indeed already shown that it is possible to use locally generated energy as a means for peak shaving and delaying the beginning of the heating season.

This work can also be extended by analyzing the economic and environmental assessment combining technologies such as hydro power, bio mass and geothermal energy that can be applicable to the Junction district of Geneva. Quantifying the level of integration of those other renewable energy scenarios can allow to get a more accurate and global model. Finally, the energy demand at the district scale can be modelized more precisely as shown by previous studies [48,92].

This study has provided the basis to a model that can be used at an early-design stage by urban planners or energy providers to obtain some key performance indicators on the energy systems they want to design. As it is modular and can obtain its inputs from multiple other software, it is thus an ideal tool to have a comprehensive assessment of urban energy systems. New planning scenarios can thus be evaluated from the demand and supply side and the implications on the reduction of the carbon and energetic footprint can be handled in one framework.

**Author Contributions:** K.S., A.T.D.P., J.-L.S. and D.M. conceived of and designed the experiments; K.S. and D.M. performed the experiments; K.S., A.T.D.P. and D.M. analyzed the data; K.S., A.T.D.P., J.-L.S. and D.M. wrote the paper.

**Funding:** This research project has been financially supported by the Swiss Innovation Agency Innosuisse and is part of the Swiss Competence Center for Energy Research SCCER FEEB&D. The APC was partially funded by the Ecole Polytechnique Fédérale de Lausanne Library.

**Acknowledgments:** The authors would like to thank the two reviewers and the editor for their valuable comments and suggestions.

**Conflicts of Interest:** The authors declare no conflict of interest. The funders had no role in the design of the study; in the collection, analyses, or interpretation of data; in the writing of the manuscript, or in the decision to publish the results.

## Abbreviations

The following abbreviations are used in this manuscript:

| | |
|---|---|
| SI-REN | Services Industriels des Energies Renouvelables de Lausanne |
| COP | Coefficient of Performance |
| $E_s$ | Energy sold to grid |
| $E_b$ | Energy bought from grid |
| $IC$ | Installation cost of the technology-tax incentives |
| $OM$ | Operations and maintenance cost of the technology |
| ST | Solar Thermal |
| SPV | Solar Photovoltaics |
| LI-ion | Lithium-Ion battery |
| LEA | Lead-Acid battery |
| O&M | Operation and maintenance cost |
| $D_t$ | Hourly demand |
| $G_t$ | Hourly generation |
| $mCO_2$ | $CO_2$ unitary emissions of a specific technology in [kg/kW] |
| $eCO_2$ | $CO_2$ total emissions from specific technology [kg/kWh] |
| $Q_c$ | Capacity of the specific technology in [kW] |
| $Q_p$ | Generation of the specific technology in [kWh] |
| LCOE | Levelized cost of energy [CHF/kWh] |
| NPV | Net present value [CHF] |
| MSC | Minimum state of charge |

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
