# Peer review of "Eco-Sim: A Parametric Tool to Evaluate the Environmental and Economic Feasibility of Decentralized Energy Systems"

_energies, doi:10.3390/en12050776_

Round 1

Reviewer 1 Report

Dear authors,

The paper aims to present Eco-Sim, a novel simulation tool foe techno-economic and environmental assessment of a distributed energy system. The work is interesting and relevant for the scientific community. Additionally, it needs some work to have enough quality for publication. The following list summarizes the comments added in the manuscript:

1) The literature survey is not enough. There is a sentence explaining that previous modelling tools "have a course temporal resolution" this is true but I have not seen an improvement with this current work. I saw that the main interest here is the environmental analysis together with the LCOE and NPV calculations.

2) The authors do not explain other simulation tools well enough. This needs to be improved considerably. There is a simulation tool called SAM that has a similar aim to this tool presented here although it is not mentioned anywhere. HOMER and BARTHome are used for validation but there are not cited in the text or either explain how they work.

3) The organization of the paper is not adequate. The "2000-Watt" society is explained by the end of the manuscript and should be introduced in the introduction section. The methodology does not include the assumptions of the NPV calculations (discount rate and years of investment period).

4) Some results are not explained enough and not compared with other tools adequately. Please, show the comparison of the simulations using the tools and extend the explanation of the results.

5) The manuscript needs an explanation of the system at the beginning of the paper, There is an schematic later in the text (page 11). Please, move this schematic to the introduction and explain it in detail. If necessary, please modify the figure to help explaining the system.

6) All figures are cut from the edges which must be modified.

7) This part " Acknowledgments: In this section you can acknowledge any support given which is not covered by the author438 contribution or funding sections. This may include administrative and technical support, or donations in kind439 (e.g., materials used for experiments)" Have to be completed.

With this improvements, the article can be submitted for a second review.

Author Response

Dear reviewer,

Thank you very much for your comments. We have really appreciated your insightful remarks and taken all of them into account by modifying consequently the manuscript. We think that the manuscript has significantly improved based on these improvements.

1)     The literature survey is not enough. There is a sentence explaining that previous modelling tools "have a coarse temporal resolution" this is true but I have not seen an improvement with this current work. I saw that the main interest here is the environmental analysis together with the LCOE and NPV calculations.

Thank you for this comment. We have considerably modified the introduction and added new references as well as explained multiple points and define the focus better.

2)     The authors do not explain other simulation tools well enough. This needs to be improved considerably. There is a simulation tool called SAM that has a similar aim to this tool presented here although it is not mentioned anywhere. HOMER and BARTHome are used for validation but there are not cited in the text or either explain how they work.

We have completed the text throughout the manuscript to include a better description of the models.

3)     The organization of the paper is not adequate. The "2000-Watt" society is explained by the end of the manuscript and should be introduced in the introduction section. The methodology does not include the assumptions of the NPV calculations (discount rate and years of investment period).

We have corrected this in the manuscript. Additionally, we had indeed missed to include the discount rates. As for the investment period, we made this clearer for the text.

4)     Some results are not explained enough and not compared with other tools adequately. Please, show the comparison of the simulations using the tools and extend the explanation of the results.

The comparison of the models have been improved and the graphics changed. Note that due to our license agreement with the industrial partner it is not possible for us to provide any other additional detail on the products (BARTHome and BARTPower).

5)     The manuscript needs an explanation of the system at the beginning of the paper, There is an schematic later in the text (page 11). Please, move this schematic to the introduction and explain it in detail. If necessary, please modify the figure to help explaining the system.

Thank you for this suggestion. We have modified the figure and its location in the manuscript.

6)     All figures are cut from the edges which must be modified.

Sorry about this. We have corrected the figures throughout the manuscript.

7)     This part " Acknowledgments: In this section you can acknowledge any support given which is not covered by the author contribution or funding sections. This may include administrative and technical support, or donations in kind (e.g., materials used for experiments)" Have to be completed.

We have modified this in the manuscript.

Reviewer 2 Report

General Comments

This paper aims to present a decision-making support tool “Eco-Sim” to review costs and environmental impacts of solar PV, battery technologies, solar thermal panels and heat pumps for urban planners, and energy system designers. This tool can be used at an early design stage to investigate the techno-economic assessment at an hourly time step throughout a typical year. This paper validated the model and extended an investigation with both thermal and electrical energy simultaneously by adding solar thermal panel and heat pumps.

I agree with the necessary of this research because it is very important to present the proposed decision-making support tool for investigating techno-economic assessment and environmental assessment for PV panels, thermal storage, and heat pumps. However, I am a bit critical about this paper for the following reasons. First, this paper is not well-organized as a whole and the differences with other researches areinsufficient. In additionI think that the manuscript should contain sufficient contributions to overcome the limitations presented in this study. For these reasons, this paper should be revised before publication.

Technical Comments

✓ The abstract section should comprise in a systematic way by representing research background, purpose, methodology, and conclusion. However, the authors did not write the abstract section systematically. So, I recommend the authors to reorganize the abstract section.

✓ In the introduction section, the literature review of previous studies is not enough. Also, the limitations of the existing studies and the difference between this study and the previous studies are not clearly presented. In particular, since the limitations of methodologies used in previous studies are not clearly indicated in the manuscript, it is considered that there is a limit to securing the validity of methodologies used in this study. Therefore, the authors should compensate this part for improving the originality and validity of this study.

✓ The authors described the methodologies regarding the techno-economic aspects and CO2 emissions (environmental aspect) quite simply. For these reasons, the overall methodology and results of this paper look very weak, and it is likely that the potential readers who interested in this field of study may have difficulty understanding the results of this study. Therefore I suggest that the authors should supplement and reorganize systematically this part.

✓ In the conclusion section, the key findings and contributions of this study should be clearly presented. However, the contribution of this study presented in the conclusion section is insufficient. Therefore, I recommend that the authors compensate the detailed descriptions on how the results of this study may contribute to the researchers, urban planners, etc.

✓ Finally, I recommend the authors to proofread the entire manuscript once again to make sure there is no awkward expressions and grammatical mistakes. Also, some graphs are too dim to read or are be cut. Therefore, it is necessary to increase the resolution of a graph (Fig. 2) and check the graphs (e.g., Fig. 3, Fig. 10, and Fig. 11) in this paper as a whole.

Author Response

Response to Reviewer #2

General Comments

This paper aims to present a decision-making support tool “Eco-Sim” to review costs and environmental impacts of solar PV, battery technologies, solar thermal panels and heat pumps for urban planners, and energy system designers. This tool can be used at an early design stage to investigate the techno-economic assessment at an hourly time step throughout a typical year. This paper validated the model and extended an investigation with both thermal and electrical energy simultaneously by adding solar thermal panel and heat pumps.

I agree with the necessary of this research because it is very important to present the proposed decision-making support tool for investigating techno-economic assessment and environmental assessment for PV panels, thermal storage, and heat pumps. However, I am a bit critical about this paper for the following reasons. First, this paper is not well-organized as a whole and the differences with other researches are insufficient. In addition, I think that the manuscript should contain sufficient contributions to overcome the limitations presented in this study. For these reasons, this paper should be revised before publication.

Dear reviewer,

Thank you very much for your remarks and suggestions. We have really appreciated your comments and have significantly modified the manuscripts based on them. We hope that you will find the manuscript has been improved. We have replied below to each of your comments individually.

Technical Comments

 The abstract section should comprise in a systematic way by representing research background, purpose, methodology, and conclusion. However, the authors did not write the abstract section systematically. So, I recommend the authors to reorganize the abstract section.

Thank you for this suggestion. We have now restructured and changed the abstract.

 In the introduction section, the literature review of previous studies is not enough. Also, the limitations of the existing studies and the difference between this study and the previous studies are not clearly presented. In particular, since the limitations of methodologies used in previous studies are not clearly indicated in the manuscript, it is considered that there is a limit to securing the validity of methodologies used in this study. Therefore, the authors should compensate this part for improving the originality and validity of this study.

We have significantly modified the introduction and added new references. We have also made it clearer what were the main improvements that the current study was bringing with regards to previous studies.

 The authors described the methodologies regarding the techno-economic aspects and CO2 emissions (environmental aspect) quite simply. For these reasons, the overall methodology and results of this paper look very weak, and it is likely that the potential readers who interested in this field of study may have difficulty understanding the results of this study. Therefore, I suggest that the authors should supplement and reorganize systematically this part

Thank you very much for this suggestion. We have restructured the Section 2, provided links to previous studies and also added new complementary information to Section 3 to make this information clearer to the reader.

 In the conclusion section, the key findings and contributions of this study should be clearly presented. However, the contribution of this study presented in the conclusion section is insufficient. Therefore, I recommend that the authors compensate the detailed descriptions on how the results of this study may contribute to the researchers, urban planners, etc.

We have added some additional comments in the conclusion to demonstrate in a better way how the results of this research can contribute to other researchers as well as to urban planners.  Note that besides the paper we will also provide on a freely available server the code and the dataset used.

 Finally, I recommend the authors to proofread the entire manuscript once again to make sure there is no awkward expressions and grammatical mistakes. Also, some graphs are too dim to read or are be cut. Therefore, it is necessary to increase the resolution of a graph (Fig. 2) and check the graphs (e.g., Fig. 3, Fig. 10, and Fig. 11) in this paper as a whole.

Thank you for this comment. We have modified the typos and other grammatical mistakes and had the manuscript proofread by a native speaker. The figures have also been modified in the text.

Round 2

Reviewer 1 Report

The authors have addressed the comments made on the article that it is ready for publication.

Reviewer 2 Report

General Comments

This paper aims to present a decision-making support tool “Eco-Sim” to review costs and environmental impacts of solar PV, battery technologies, solar thermal panels and heat pumps for urban planners, and energy system designers. This tool can be used at an early design stage to investigate the techno-economic assessment at an hourly time step throughout a typical year. This paper validated the model and extended an investigation with both thermal and electrical energy simultaneously by adding solar thermal panel and heat pumps.

Overall, the framework of this paper is well organized with solid method and data support. Compared to the first draft of this manuscript, I think the authors have faithfully revised this paper based on reviewer’s comments. Therefore, this paper can be accepted for the publication.

Technical Comments

✓ The authors have revised the abstract section considering readers as reviewer’s comments. The abstract section have revised in a systematic way by representing research background, purpose, methodology, and conclusion as reviewer’s commentsThis paper explained the abstract section more systematically.

✓ In the introduction section, the authors have added the literature review of previous studies enough. Also, as reviewer’s comment, the limitations of the previousstudies and the difference between this study and the previous studies are presented in the methodology sectionHowever, I recommend that the authors explained the difference between this study and the previous studies should be explained in the introduction section. Therefore, it is considered that there is not difficult for readers to understand the process of this study. Therefore, the authors should compensate this part for understanding of this study.

✓ The authors described the methodologies regarding the techno-economic aspects and CO2 emissions (environmental aspect) well. As reviewer’s comments, methodologies and results in this paper does not look very weak. This paper explained the methodology and results section more adequately.

✓ In the conclusion section, as reviewer’s comments, the key findings and contributions of this study is clearly presented. Also, the authors revised the contribution of this paper sufficient.

✓ Overall, I think the authors proofread the entire manuscript well and there is no awkward expressions and grammatical mistakes.